# WHO'S YOUR JUDGE? ON THE DETECTABILITY OF LLM-GENERATED JUDGMENTS

## ABSTRACT

Large Language Model (LLM)-based judgments leverage powerful LLMs to efficiently evaluate candidate content and provide judgment scores. However, the inherent biases and vulnerabilities of LLM-generated judgments raise concerns, underscoring the urgent need for distinguishing them in sensitive scenarios like academic peer reviewing. In this work, we propose and formalize the task of judgment detection and systematically investigate the detectability of LLM-generated judgments. Unlike LLM-generated text detection, judgment detection relies solely on judgment scores and candidates, reflecting real-world scenarios where textual feedback is often unavailable in the detection process. Our preliminary analysis shows that existing LLM-generated text detection methods perform poorly given their incapability to capture the interaction between judgment scores and candidate content—an aspect crucial for effective judgment detection. Inspired by this, we introduce *J-Detector*, a lightweight and transparent neural detector augmented with explicitly extracted linguistic and LLM-enhanced features to link LLM judges' biases with candidates' properties for accurate detection. Experiments across diverse datasets demonstrate the effectiveness of *J-Detector* and show how its interpretability enables quantifying biases in LLM judges. Finally, we analyze key factors affecting the detectability of LLM-generated judgments and validate the practical utility of judgment detection in real-world scenarios.

## 1 INTRODUCTION

Taking advantage of the powerful Large Language Models (LLMs), the paradigm of LLM-based judgment (Zheng et al., 2023; Li et al., 2024) has been proposed, designed to automate and scale up various annotation and reviewing applications (Lee et al.; Zhu et al., 2025). By combining powerful LLMs with well-designed prompting strategies, LLM-based judgment enables human-like evaluation of long-form and open-ended generation in a more cost-efficient manner. For example, LLM-based judgment has been increasingly used in the peer review of leading AI conferences (Liang et al., 2024).

Despite this remarkable progress, many recent studies point out various biases of LLM-generated judgment toward spurious features, such as length and affinity (Ye et al., 2024; Li et al., 2025a). Besides, the vulnerability of the LLM judgment system has also been revealed, that several maliciously-designed and hard-to-detect tokens or words can fool the LLM judges to give much inconsistent scores despite the candidates' genuine quality (Shi et al., 2024; Zhao et al., 2025). Recently, in the scenario of academic peer reviewing, some researchers sneak hidden prompts, which are invisible and usually presented as a white font on a white background, into their papers to instruct LLMs to only provide positive feedback and thus trick AI reviewers[1]. All these challenges highlight the importance of distinguishing LLM-generated judgments to guarantee the assessment's fairness and reliability.

To address this concern, we propose the judgment detection task, which aims at examining the detectability of LLM-generated judgments across diverse scenarios. Unlike existing machine-generated text detection task that focuses on textual content (Mitchell et al., 2023), judgment detection targets at distinguishing LLM-generated from human-produced judgments solely based on the *candidate content* and *judgment scores* (as illustrated in Figure 1). For instance, in academic paper reviewing,

---

[1] https://www.theregister.com/2025/07/07/scholars_try_to_fool_llm_reviewers/

judgment detection will be performed using only the candidate paper and its assigned ratings (e.g., soundness, novelty, overall score), without accessing the full review text. This setting is particularly important for real-world scenarios where textual feedback is often unavailable in the detection process. For example, reviewers who adopt AI-generated reviews may intentionally submit minimal textual content, such as "N/A" to evade detection. Moreover, in the evaluation data labeling scenario, annotators are typically required to provide only the judgment scores. Score-based judgment detection is especially critical in these scenarios to identify the illegal use of LLM-generated judgment and guarantee assessment reliability.

Developing a good LLM-generated judgment detector is not trivial. In our warm-up analysis, we identify two key types of information for judgment detection which are not jointly considered in existing related approaches: ❶ **Judgment-Intrinsic Features**, which capture patterns within the judgment score distribution, and ❷ **Judgment-Candidate Interaction Features**, which capture the interaction between judgment scores and candidate content. Building on them, we find that existing LLM-generated text detection methods fail to capture

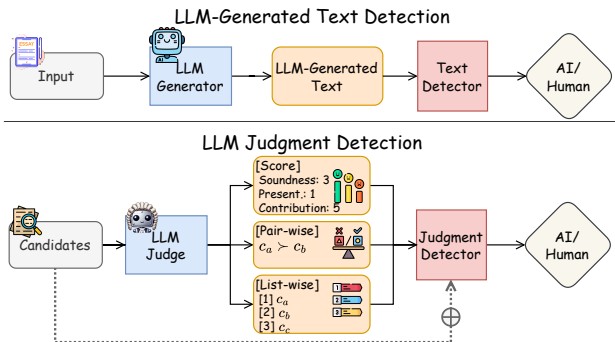

Figure 1: Comparison between LLM-generated judgment detection and text detection.

Judgment-Candidate Interaction Features, leading to subpar performance—especially in single-dimension settings, where each judgment consists of a single score assessing one aspect of the candidates. To address this, we introduce *J-Detector*, a lightweight and interpretable neural detector designed specifically for LLM-generated judgment detection. *J-Detector* is augmented with explicitly extracted linguistic and LLM-enhanced features to capture systematic correlations between judgment scores and candidate features that LLM judges are often biased toward, thereby effectively leveraging these biases for more accurate detection.

Experiments across diverse judgment datasets demonstrate the effectiveness of *J-Detector* and the two types of augmented features. Besides, we showcase how to leverage the interpretability of *J-Detector* to enable bias quantification in LLM judges. Finally, we analyze key factors affecting the detectability of LLM-generated judgments and demonstrate a real-world application that integrates judgment detection with text-based detection to identify AI-generated reviews in an academic peer reviewing scenario. In summary, our key contributions are:

- We propose, for the first time, the judgment detection task, which aims at distinguishing human and LLM judgments based on judgment scores and candidate content.
- We design *J-Detector*, a lightweight and interpretable detection method, that effectively bridges candidate and judgment information with linguistic and LLM-enhanced features.
- Through extensive experiments, we demonstrate the advantages of *J-Detector*, identify key factors driving judgment detectability, and show the utility of judgment detection in real-world applications.

## 2 RELATED WORK

**LLM-as-a-judge**, first introduced by Zheng et al. (2023), leverages LLMs to automatically evaluate candidate content and assign scores as judgment results. This paradigm has been expanded to diverse applications to judge various types of candidates, including paper quality assessing (Jin et al., 2024), document relevance measurement (Rahmani et al., 2024), and reasoning trace correctness verification (Zhang et al.), driving substantial progress in automatic assessment. Despite these advances, recent studies highlight notable limitations. Research has uncovered systematic biases in LLM-generated judgments, where evaluations are influenced by spurious features such as response length or superficial affinity rather than genuine content quality (Ye et al., 2024; Li et al., 2025a). Moreover, adversarial work demonstrates that LLM judges can be manipulated with a few carefully crafted, hard-to-detect tokens or phrases, which induce disproportionately high scores misaligned with actual candidate quality (Shi et al., 2024; Zhao et al., 2025). To mitigate these issues, methods

such as bias quantification (Ye et al., 2024) and human-in-the-loop calibration (Wang et al., 2023a) have been proposed. Building on this line of research, we introduce a new task, judgment detection, that aims to distinguish and prevent the misuse of LLM-generated judgments.

**AI-generated Text Detection** aims to distinguish machine-generated from human-produced text, evolving from early stylometric and perplexity-based methods (Gehrmann et al., 2019; Zellers et al., 2019) to supervised classifiers (Ippolito et al., 2020; Mitchell et al., 2023), and more recently toward robust, generalizable approaches such as zero-shot prompting and watermarking (Sun & Lv, 2025; Mao et al., 2025). Another relevant line of work for us is the detection of LLM-generated peer reviews (Tao et al.; Yu et al.; Rao et al., 2025), where detectors are designed to distinguish machine-written reviews from human-authored ones. However, these approaches rely on textual review content, which is often unavailable in broader judgment settings. In this work, we borrow insights from both fields and propose judgment detection to explore the detectability of LLM-produced judgment, using judgment scores without accessing textual feedback.

## 3 TASK STATEMENT

A *judgment* refers to an assessment made over one or more candidates $c \in \mathcal{C}$, where $|\mathcal{C}|$ denotes the size of the candidate set. A judgment score is denoted by $j = (j_1, \ldots, j_d) \in \mathcal{Y}^d$. It can be either *single-dimensional* ($d = 1$), reflecting an assessment toward a single aspect, or *multi-dimensional* ($d > 1$), where each component $J_i$ corresponds to a distinct evaluation aspect (e.g., relevance, fluency, coherence). With these definitions, we formulate the task as follows:

**Definition 3.1** (Judgment Detection). LLM-generated judgment detection is defined over *judgment groups*. A judgment group is given by $G = \{(c^i, j^i)\}_{i=1}^k$, where each candidate $c^i \in \mathcal{C}$ is paired with a judgment score $j^i \in \mathcal{J}$. The task is to classify whether a group $G$ originates from a human judge or from an LLM. Formally, the label space is $L = \{0, 1\}$, where $\ell = 0$ denotes human-produced judgments and $\ell = 1$ denotes LLM-generated judgments. The goal is to learn a function $f_\theta : G \to [0, 1]$, where $f_\theta(G)$ outputs the probability that $G$ was generated by an LLM. The final prediction is obtained as $\hat{y} = \mathbb{I}[f_\theta(G) \geq \tau]$, with threshold $\tau \in [0, 1]$ and indicator function $\mathbb{I}[\cdot]$.

When the group size is 1, *i.e.*, $|G| = 1$, the task is degraded to an i.i.d. (instance-level) detection setting, where each judgment is treated independently. When $|G| > 1$, the group setting better reflects real practice, since judgments are usually produced in batches (e.g., a reviewer scores multiple papers or an annotator evaluates a set of model outputs), and collective patterns across the group can reveal whether the judgments are human-produced or LLM-generated.

## 4 WARM-UP ANALYSIS: WHAT MATTERS FOR LLM-GENERATED JUDGMENT DETECTION?

To understand the key ingredients of a reliable judgment detector, we first conduct a warm-up study by adapting LLM-generated text detection methods to the judgment detection setting. Specifically, we employ small language models (SLM)-based detectors (Wu et al., 2024), *RoBERTa* and *Longformer*, as $f_\theta$ and evaluate them on four datasets: *Helpsteer2*, *Helpsteer3*, *NeurIPS*,

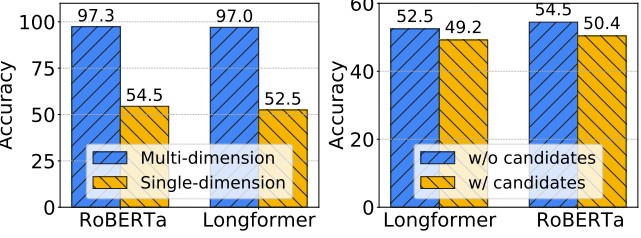

(a) Multi- vs Single-dimension   (b) Can. on Single-dimension

Figure 2: Multi- vs Single-dimension and Candidate Effect.

and *ANTIQUE*. More information about implementation and dataset can be found in Section 6.1.

**Multi-dimension vs Single-dimension performance.** As shown in Figure 2 (a), both RoBERTa and Longformer achieve high accuracy in the *multi-dimension* scenarios (Helpsteer2 and NeurIPS) but perform poorly in the *single-dimension* scenarios (Helpsteer3 and ANTIQUE). We assume that this discrepancy arises because, in multi-dimension settings, the detectors can exploit distributional differences in how humans and LLMs assign scores across multiple judgment dimensions, whereas in single-dimensional settings, such distributional cues are almost absent.

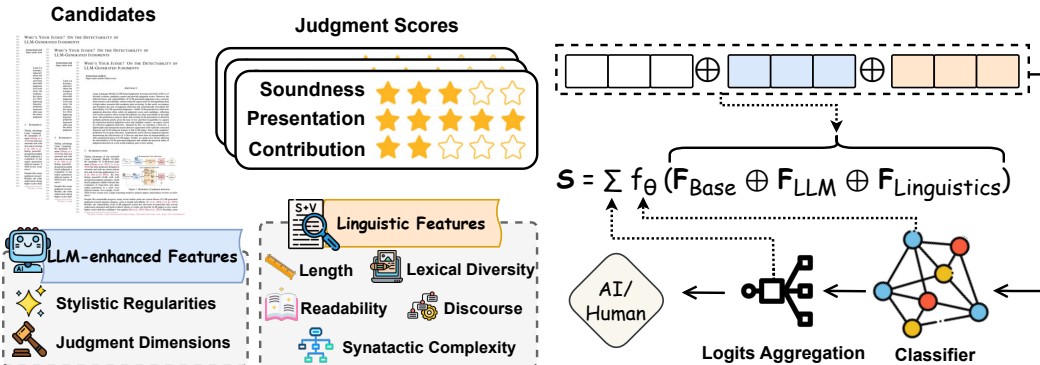

Figure 3: The overview pipeline of our *J-Detector* for LLM-generated judgment detection.

**Adding candidate information.** We further extend the single-dimension setting by providing candidate texts alongside their judgments, exploring whether the detectors can extract and leverage judgment–candidate interaction information. As shown in Figure 2 (b), however, adding candidates does not lead to any performance improvement. This suggests that SLM-based detectors are unable to directly capture and utilize the interaction between judgments and candidate content from raw input.

**Takeaway.** From this warm-up study, we identify two complementary types of information that a reliable judgment detector should exploit: ❶ **Judgment-Intrinsic Features**, revealed by the large performance gap between multi-dimension and single-dimension settings, indicating that distributional patterns within judgment scores themselves can be highly informative; and ❷ **Judgment-Candidate Interaction Features**, which capture how judgment scores relate to the underlying candidate content but remain largely unexplored by existing methods. These findings highlight that existing SLM-based text detection methods mainly leverage judgment-intrinsic patterns but fail to capture judgment–candidate interactions, which are especially critical in single-dimension scenarios.

## 5  *J-Detector*: A LIGHTWEIGHT AND INTERPRETABLE DETECTOR

To address the limitation of existing text detectors and design an effective and robust approach for LLM-generated judgment detection, we first identify three criteria that a good LLM-generated judgment detector should embody:

- (**Accurate**) The detection method should be able to leverage both Judgment-Intrinsic Features and Judgment-Candidate Interaction Features to deliver reliable detection results in various scenarios.
- (**Efficient**) Both the training and inference of the detector should incur minimal computational overhead, enabling the method to be deployed in large-scale judgment detection scenarios.
- (**Interpretable**) The detection method should be interpretable to support bias analysis in LLM judges.

Following these principles, we design *J-Detector*, an accurate, lightweight and interpretable detector involving the following components. The overview pipeline is presented in Figure 3.

**Feature Augmentation.** Let $\mathbf{F}$ denote the instance-level feature vector used by *J-Detector*. We construct it by concatenating three types of features together:

$$\mathbf{F} = \mathbf{F}_{\text{base}} \oplus \mathbf{F}_{\text{LLM}} \oplus \mathbf{F}_{\text{linguistic}}, \tag{1}$$

where $\mathbf{F}_{\text{base}}$ contains the *given judgment scores*. $\mathbf{F}_{\text{LLM}}$ and $\mathbf{F}_{\text{linguistic}}$ are *LLM-enhanced features* and *linguistic features* we extract from candidates content, which act as distilled information of candidates and are leveraged to link judgment scores with candidates' content.

LLM-enhanced Features. Borrowing insights from LLM-based text detection methods (Bao et al., 2024), we propose LLM-enhanced features to produce the following types of features:

- *Stylistic regularities:* scores reflecting surface polish and presentation patterns of the candidates, including *style*, *wording*, and *format*. These aim to capture the spurious preference LLM judges tend to have over superficial attributes (Li et al., 2025a).

- *Judgment-aligned dimensions:* scores aligned to the same dimensions used in the given judgment scores. These aim to enhance features by leveraging the similarity of biases across LLM judges.

By injecting these high-level, bias-informed signals, LLM-enhanced Features enable the detector to better capture subtle judgment patterns that are difficult to learn from raw candidate content alone.

Linguistic Features. We further introduce linguistic features $\mathbf{F}_{\text{linguistic}}$ to capture low-level linguistic regularities that often correlate with systematic biases of LLM judges. Specifically, we extract the following aggregated features from the candidate content:

- *Length:* total token and character counts, as well as average sentence length, to capture the *length bias* where LLM judges favor lengthy content and responses (Wei et al.).
- *Lexical diversity:* unique-token ratio and average word length, which reflect the *surface beauty bias* of LLM-generated judgments compared to human-produced ones (Chen et al., 2024).
- *Readability:* a composite readability index (e.g., Coleman–Liau), measuring the *fluency bias* where LLMs tend to favor superficially fluent texts, disregarding their true quality (Wu & Aji, 2025).
- *Syntactic complexity:* dependency tree depth and average dependency distance, used to identify the *complexity bias* often observed in LLM judges (Ye et al., 2024).
- *Discourse/hedging:* the frequency of discourse markers and hedging expressions, capturing the *presentation bias* of LLM, which prefer content with confident tones (Kharchenko et al., 2025).

These features provide a compact yet informative summary of linguistic cues, enabling the detector to exploit stable and interpretable signals that are complementary to LLM-enhanced features.

**Model Training.** Given labeled instances $(\mathbf{F}, y)$, we train a lightweight binary classifier $f_\theta$ (e.g., RandomForest (Breiman, 2001)) to output a *logit* $z \in \mathbb{R}$ indicating the likelihood that the judgment was generated by an LLM ($y = 1$) or by a human ($y = 0$). The classifier is trained using the augmented feature $\mathbf{F}$ and serves as the instance-level building block for group-level decisions.

**Group-level Aggregation.** To enable the group-level detection setting, we propose a simple aggregation method to produce the group-level label give each single prediction. Given a group $G$ consisting of $k$ judgments with instance-level logits $\{\hat{z}_1, \ldots, \hat{z}_k\}$, we aggregate the evidence using sum aggregation: $\text{score}(G) = \sum_{i=1}^{k} \hat{z}_i$.

In summary, *J-Detector* is designed to satisfy the three criteria identified at the beginning of this section. First, by incorporating both LLM-enhanced and linguistic features, it is able to capture not only Judgment-Intrinsic Features but also critical Judgment–Candidate Interaction Features, enabling accurate detection across single-dimensional and multi-dimensional scenarios. Second, it builds on a lightweight binary classifier, making both training and inference highly efficient and thus suitable for large-scale deployment. Third, since the features are semantically clear and the classifier itself is simple, the framework offers strong interpretability, which can be leveraged to systematically quantify and analyze the biases of LLM judges.

## 6 MAIN EXPERIMENT

### 6.1 EXPERIMENT SETTINGS

**Datasets.** We build a comprehensive LLM-generated judgment detection dataset, *JD-Bench*, which integrates four representative datasets covering three judgment types: pointwise, pairwise and list-wise (Li et al., 2024). Among them, *HelpSteer2* provides large-scale pointwise human ratings of LLM responses for helpfulness evaluation, while *HelpSteer3* extends this with pairwise human preference comparisons. The *NeurIPS Review dataset* offers expert peer reviews with multi-dimensional scores such as soundness and novelty, representing high-stakes evaluation. Finally, *ANTIQUE* supplies listwise human judgments for ranking documents in non-factoid question answering. All four datasets contain human-labeled judgments as reliable references, and we further collect LLM-generated judgments from a diverse pool of models. In total, *JD-Bench* covers a wide spectrum of model families, including *OpenAI*, *Anthropic*, and *Google* for closed-source models, and *LLaMA*, *Qwen*, *Mistral*, and *DeepSeek* for open-source models, ensuring diversity in judgment patterns.

**Compared Methods.** In our main experiment, we compare our proposed *J-Detector* against a series of baseline methods, all of which are listed as follows:

- **SLM-based Detector.** In line with SLM-based text detectors (Yu et al., 2025), this approach feeds either the judgment scores alone or the judgment scores together with the candidate content (w/ candidates) to train a small language model-based classifier to predict whether the judgment was produced by a human or from an LLM.
- **LLM-as-a-judge-detector.** Inspired by logits-based detection in AI-generated text detection (Mitchell et al., 2023), where a surrogate LLM is used to compute likelihoods, we adopt a single LLM that first generates judgment scores and then compares them with the judgment scores to be detected, making the detection decision based on their similarity.
- **Sample-level LLM-based Analysis.** Inspired by recent agent-based frameworks that maintain guideline banks for distinguishing human and AI text (Li et al., 2025b), we let the LLM analyze Human–LLM judgment-candidate pairs to extract concise instance-level features (e.g., length bias in LLM judgments), which are stored in a feature bank to capture regularities useful for detection.
- **Distribution-level LLM-based Analysis.** Drawing inspiration from recent work that guides LLMs in structured extraction and analysis of visual summaries (Liu et al., 2025), we provide the model with dataset-level summaries (e.g., per-label histograms and correlations), enabling it to incorporate global and distributional cues into the detection decision.

**Implementation Details.** We implement our *J-Detector* using three models from the Scikit-learn library (Pedregosa et al., 2011): LGBM (Ke et al., 2017), RandomForest (Breiman, 2001), and XGB (Chen & Guestrin, 2016). We employ *Qwen-3-8B* for both feature augmentation and as the backbone for LLM-based baselines. For SLM-based methods, we use *RoBERTa-base* and *Longformer-4096*. For SLM training, we use a batch size of 8 and fine-tune the SLM for 3 epochs on each dataset. In the main experiments, the group size is fixed to $k=4$. More details, including the *JD-Bench* construction, design of baseline methods, and implementation specifics are provided in Appendix B.

## 6.2 MAIN RESULT

Table 1: Main experimental results on *JD-Bench*. We report F1 and AUROC scores, with the best results highlighted in bold. Each experiment is repeated five times, and average scores are reported.

| Method | Helpsteer2 | | Helpsteer3 | | NeurIPS | | ANTIQUE | | AVG | |
|---|---|---|---|---|---|---|---|---|---|---|
| | F1 | AUROC | F1 | AUROC | F1 | AUROC | F1 | AUROC | F1 | AUROC |
| *SLM-based methods* | | | | | | | | | | |
| RoBERTa | 98.1 | 99.6 | 50.9 | 64.5 | 96.2 | 99.4 | 30.0 | 56.8 | 68.8 | 80.1 |
| RoBERTa w/ candidates | 98.1 | 99.6 | 50.0 | 63.4 | 96.3 | 99.3 | 27.6 | 56.6 | 68.0 | 79.7 |
| Longformer | 98.1 | 99.7 | 54.5 | 65.7 | 96.2 | 99.5 | 30.6 | 56.6 | 69.9 | 80.4 |
| Longformer w/ candidates | 98.1 | 99.7 | 51.4 | 64.3 | 96.2 | 99.4 | 21.8 | 48.8 | 66.9 | 78.0 |
| *LLM-based methods* | | | | | | | | | | |
| LLM | 51.5 | 50.3 | 50.3 | 50.1 | 43.9 | 50.2 | 49.6 | 49.9 | 48.8 | 50.1 |
| LLM w/ Sample-level | 49.8 | 49.7 | 49.6 | 50.2 | 50.5 | 50.4 | 50.9 | 50.3 | 50.2 | 50.2 |
| LLM w/ Distribution-level | 52.1 | 50.0 | 48.8 | 50.3 | 49.6 | 49.8 | 50.7 | 50.1 | 50.3 | 50.1 |
| LLM w/ Sample-level + Distribution-level | 58.7 | 50.4 | 49.4 | 49.6 | 51.2 | 50.2 | 50.2 | 49.9 | 52.4 | 50.0 |
| *J-Detector (ours)* | | | | | | | | | | |
| LGBM | 99.6 | **100.0** | 68.1 | 73.3 | **98.7** | **99.9** | **85.4** | **93.3** | 88.0 | 91.6 |
| RandomForest | 99.5 | **100.0** | **74.0** | **77.0** | 97.0 | 99.7 | 82.6 | 90.6 | **88.3** | **91.8** |
| XGB | **99.8** | **100.0** | 68.5 | 73.6 | 98.4 | 99.8 | 84.2 | 92.3 | 87.7 | 91.4 |

**SLM-based Methods Analysis.** As we discussed in Section 4, SLM-based methods perform strongly on multi-dimensional datasets like Helpsteer2 (98.1% F1 on RoBERTa) and NeurIPS (96.2% on RoBERTa), but drop to around 50–55% F1 on single-dimensional datasets like Helpsteer3 and Antique. Even adding candidates barely helps. This shows SLMs rely on inter-dimension patterns and fail to link judgments with candidates when such distributional cues are absent.

**LLM-based Methods Analysis.** Furthermore, all LLM-based methods hover near 50% F1 score across datasets, indicating almost random guessing. When combining with sample-level comparative analysis and distribution-level chart reasoning, LLM-based detection methods yield some gains in multi-dimensional datasets (e.g., from 51.5% to 58.7% F1 score). While this improvement doesn't appear in Helpsteer3 and ANTIQUE, we conclude that LLM-based detectors also suffer from leveraging judgments-candidates interaction, with either sample- or distribution-level methods.

**J-Detector Analysis.** Compared with them, *J-Detector* achieves the best detection performance across all 4 datasets and 2 metrics, far surpassing all baselines. Noted that in the single-dimensional judgment scenarios, *J-Detector* yields much better detection performance compared with other baselines. This demonstrates that explicitly modeling the distributional patterns and biases of LLM

judgments is crucial for accurate detection, enabling robust performance in both single-dimensional and multi-dimensional judgment detection scenarios.

**Ablation Study.** Figure 4 shows that both LLM-enhanced and linguistic features consistently improve performance across all group sizes. Removing either feature causes the F1 score to drop at every group size—for example, at k = 16, removing linguistic features lowers F1 by 5.3%, and removing both leads to a 12.3% drop. This demonstrates that the two augmented features are complementary and beneficial across all datasets and group-size settings.

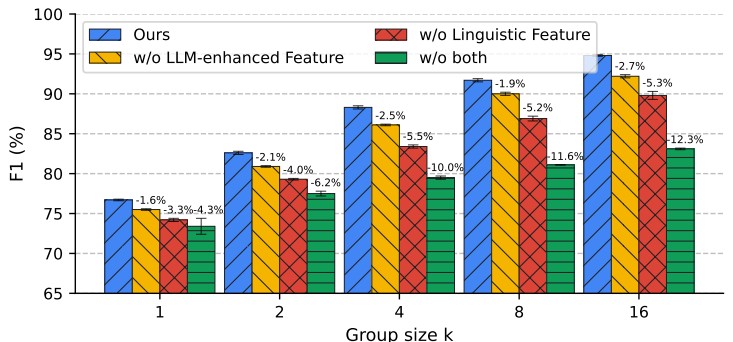

Figure 4: Ablation study on LLM-enhanced and linguistic features.

**Bias Quantification with J-Detector.** Additionally, we illustrate how the transparency and interpretability of *J-Detector* can be leveraged to quantify biases in LLM-as-a-judge by analyzing which features most strongly influence the detector's decisions. Specifically, we select the top 20 most important features ranked by their absolute coefficient values, and report the results on the Helpsteer2 and NeurIPS datasets in Figure 5. The analysis reveals that base judgment score features provide strong signals for distinguishing LLM-generated judgments from human-produced ones, highlighting the critical role of *Judgment-Intrinsic Features*. As shown in the figure, LLM judges exhibit the strongest bias in the *complexity* and *confidence* dimensions for the two datasets, respectively, consistent with prior findings that LLMs tend to favor more complex responses (Ye et al., 2024) and often display overconfidence (Kadavath et al., 2022). In addition, we observe common cross-dataset biases such as *length bias* (captured by `average_dependency_length`) and *beauty bias* (reflected in style-related scores), which echo broader concerns about spurious preference and correlations in LLM-based judgments (Wang et al., 2023b; Shi et al., 2024).

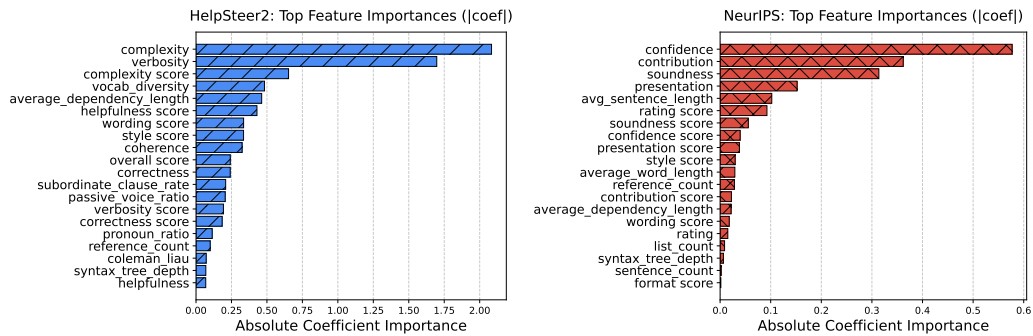

Figure 5: LLM-as-a-judge bias quantification on Helpsteer2 and NeurIPS.

# 7 FURTHER ANALYSIS

In this section, we empirically analyze the key factors that influence the detectability of the LLM-generated judgment, as well as present a real-world application to combine LLM-based judgment detection with text detection in real-world academic peer reviewing scenarios.

## 7.1 DETECTABILITY ANALYSIS

**Detectability analysis across group size, judgment dimensions, and rating scale.** Figure 6 shows that group size is a key factor in the detectability of LLM-generated judgments: the F1 score consistently improves as the group size increases across all four datasets (e.g., F1 score in Helpsteer3 rises from 63.9% at $k = 1$ to 85.0% at $k = 16$). The number of judgment dimensions also plays

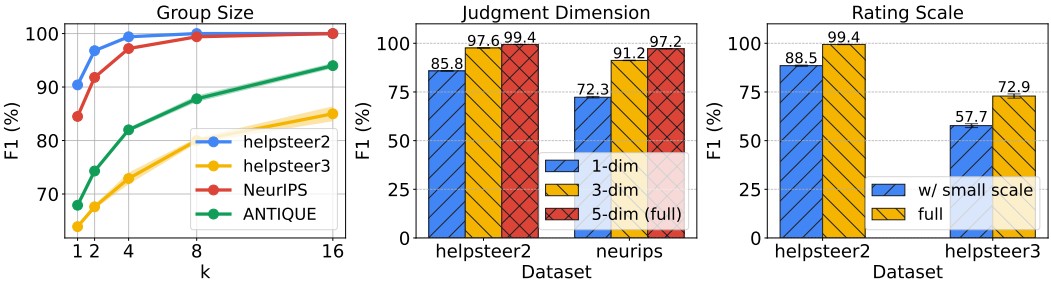

Figure 6: Detectability analysis on group size, judgment dimensions and rating scale.

an important role; for instance, when only a single dimension out of the five is used in the NeurIPS dataset, the F1 score drops substantially (from 97.2% to 72.3%). This confirms that multi-dimensional judgments provide richer distributional signals as Judgment-Intrinsic Features for detection.

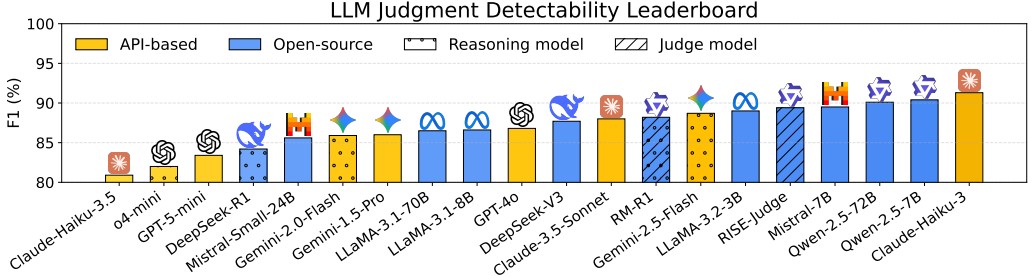

Figure 7: Detectability leaderboard on 20 LLMs. RM-R1 and RISE-Judge are based on Qwen-2.5-7B.

In addition, the granularity of the rating scale further impacts detectability: collapsing to a coarse scale (e.g., merging $-3/-2/-1$ into $-1$ and $1/2/3$ into 1 in Helpsteer3) leads to degraded performance (e.g., F1 drops from 72.9% to 57.7%). Overall, these results underscore that group size, the number of dimensions, and the rating scale collectively shape how detectable LLM-generated judgments are.

**Detectability of Various LLM Judges.** Additionally, Figure 7 summarizes the detectability leaderboard across 20 LLMs, averaged over different group sizes. We observe that API-based models (yellow bars) are generally more difficult to detect than open-source models (blue bars), indicating that closed commercial systems such as

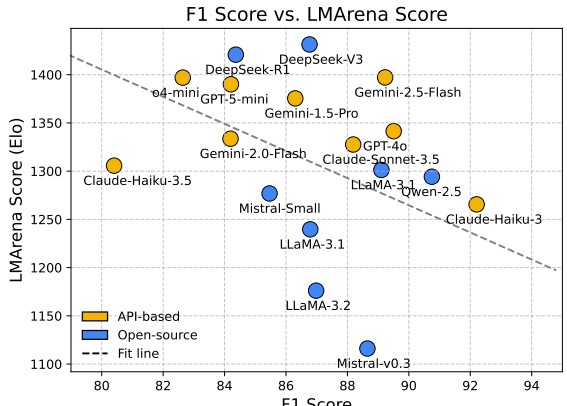

Figure 8: Correlation between judge LLMs' detectability and LMArena score.

GPT-5-mini and Claude-Haiku-3 produce judgments that more closely resemble human annotations.

Within the same model families, larger models tend to be less detectable than smaller ones: for instance, among LLaMA-3 and Qwen-2.5 families, larger models consistently achieve lower detectability. Moreover, reasoning models (dotted bars) and specialized judge models (striped bars) consistently achieve higher robustness than standard LLMs, suggesting that models explicitly optimized for reasoning or evaluation align more closely with human judgment distributions and are therefore harder to distinguish from human judges.

As presented in Figure 8, we also study the correlation between the detectability of different LLM judges and their LMArena score (Chiang et al., 2024), which is a proxy of LLMs' alignment degree with human preference and value. We find a clear negative correlation: models with higher

alignment scores are systematically less detectable. This observation reinforces our previous findings, supporting the hypothesis that as models become better aligned with human values, the gap between their judgments and human annotations narrows, making their outputs increasingly difficult to distinguish from those of human judges.

For LLM-generated judgment detectability, we also theoretically prove and demonstrate each influence factor's effect and put it in Appendix C.

### 7.2 JUDGMENT DETECTION WITH MULTIPLE LLM JUDGES

In this section, we examine how the detectability of LLM-generated judgments changes when multiple LLM judges are involved. This setting reflects real-world scenarios where judgments may come from a diverse pool of LLMs. As shown in Figure 9, we randomly sample 2, 3, 5, or 10 LLMs from our *JD-Bench* and mix their judgments in both the training and testing sets. We observe a substantial drop in detection performance across all four datasets (e.g., the F1 score decreases from 99.8% to 66.9% on Helpsteer2). This suggests that detecting LLM-generated judgments becomes significantly more challenging when multiple LLM judges are present, as detectors must learn to recognize distinct patterns from different models. Notably, the performance drop is relatively small on the NeurIPS dataset, indicating stronger shared biases among LLM judges in

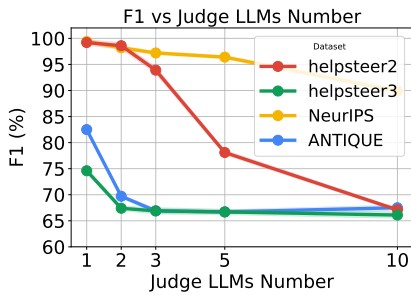

Figure 9: Detectability of LLM-generated judgment in multiple LLM judges setting.

that domain. One promising direction for future work is to explore effective LLM-generated judgment detection methods under multiple judges' settings.

### 7.3 JUDGMENT-TEXT CO-DETECTION: AN APPLICATION

In this section, we explore two real-world scenarios where LLM-generated judgment detection can support peer review authenticity checking. First, the few-shot detection setting simulates cases where a new conference is launched or the review form has changed. Here, we set the number of training samples to be 60. Second, the missing-text detection setting addresses the common case where reviews lack enough textual feedback. We simulate this setting by masking 15% of the text reviews.

Table 2: An application to leverage judgment and text feedback for AI-generated review detection in few-shot and missing review scenarios.

| Method | Few-shot | Missing review |
|---|---|---|
| w/ RoBERTa-text | 67.2 | 90.5 |
| w/ *J-Detector* | 64.4 | 86.2 |
| w/ RoBERTa-text & *J-Detector* | **74.6** | **99.3** |

The results in Table 2 show that combining the *J-Detector* with a text-based detector (RoBERTa-text) achieves the best performance in both settings (74.6% vs. 67.2% in few-shot, and 99.3% vs. 90.5% in missing-text), outperforming either method alone. This demonstrates that LLM-generated judgment detection provides complementary signals to text-based detectors and is highly valuable in real-world low-resource or judgment score-only scenarios for robust and reliable detection.

## 8 CONCLUSION

In this work we introduced judgment detection as the task of distinguishing human from LLM-generated judgments and proposed *J-Detector*, a lightweight, interpretable detector enhanced with linguistic and LLM-based features. Experiments on *JD-Bench* show that *J-Detector* consistently outperforms baselines, while our theoretical and empirical analyses reveal that detectability improves with larger group size, richer dimensions, finer rating scales, and greater human–LLM divergence. Using *J-Detector*'s transparency, we further quantified systematic biases in LLM judges, such as complexity, confidence, and length biases, and demonstrated practical value in peer-review authenticity checking. These findings establish LLM-generated judgment detection as a key safeguard for ensuring fairness and accountability in LLM-as-a-judge systems.

ETHICS STATEMENT

We adhere to the ICLR Code of Ethics. No private, sensitive, or personally identifiable data are involved. Our work does not raise foreseeable ethical concerns or produce harmful societal outcomes.

REPRODUCIBILITY STATEMENT

Reproducibility is central to our work. All datasets used in our experiments are standard benchmarks that are publicly available. We provide full details of the training setup, model architectures, and evaluation metrics in the main paper and appendix. Upon acceptance, we will release our codebase, including scripts for preprocessing, training, and evaluation, along with configuration files and documentation to facilitate exact reproduction of our results. Random seeds and hyperparameters will also be included to further ensure reproducibility.

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

CONTENTS OF THE APPENDIX

# A  THE USE OF LLMS FOR WRITING

We employed Google's Gemini 2.5 Pro and OpenAI's GPT-5 as writing assistance tools during the preparation of this manuscript. Their role was exclusively for language refinement, such as improving readability and rephrasing for clarity in an academic writing style. This usage aligns with standard academic practices for language polishing.

# B  EXPERIMENT IMPLEMENTATION DETAILS

## B.1  DETAILED DEFINITION OF VARIOUS JUDGMENT TYPES

Depending on the evaluation protocol, judgments can take multiple forms (Li et al., 2024): (i) *Score-based judgments*: $j \in \mathbb{R}$, such as a numerical rating on one or several dimensions; (ii) *Pairwise judgments*: $j \in \{(c_a \succ c_b), (c_b \succ c_a)\}$, indicating a preference between two candidates $c_a, c_b \in \mathcal{C}$; (iii) *Listwise judgments*: $j \in \pi(\mathcal{C})$, representing a permutation (ranking) $\pi$ over a candidate set.

## B.2  *JD-Bench Details*

To systematically study the detectability of LLM-generated judgments, we introduce **JD-Bench**, a large-scale benchmark that integrates diverse applications, judgment types, and model sources. JD-Bench provides a unified testbed for evaluating both existing and newly proposed detectors under realistic settings.

**Dataset Selection.**  We construct JD-Bench by aggregating data from multiple domains and judgment types, ensuring broad coverage of evaluation practices:

- **HelpSteer2** (Wang et al., 2024): HelpSteer2 is an open-source dataset designed to train and evaluate reward models for helpfulness assessment of LLM-generated responses. It contains large-scale human-annotated pointwise judgments that assign numerical scores to responses across diverse instruction-following tasks. The dataset covers multiple domains and languages, enabling robust generalization of reward models. Its fine-grained annotations make it a strong benchmark for pointwise/score-based evaluation.
- **HelpSteer3** (Wang et al., 2025): HelpSteer3 extends HelpSteer2 by collecting pairwise human preference data on LLM responses. Instead of absolute scores, annotators compare two candidate responses to the same prompt and indicate which is better, yielding high-quality comparative judgments. The dataset spans a wide range of tasks and languages, supporting cross-lingual preference modeling and fine-grained ranking evaluation.
- **NeurIPS Review Dataset** (Yu et al., 2025): This dataset comprises a large collection of real academic peer reviews from the NeurIPS conference, annotated with multi-dimensional scores such as soundness, novelty, clarity, and overall rating. It represents a domain where judgments are structured, multi-faceted, and highly consequential. The dataset captures nuanced reviewing language and decision rationales, providing a challenging benchmark for modeling human-like expert evaluation. It is especially valuable for studying judgment behavior in formal and high-stakes settings.
- **ANTIQUE** (Hashemi et al., 2020): ANTIQUE is a benchmark for non-factoid question answering, focused on ranking passages based on their relevance to user queries. It includes listwise relevance judgments collected from crowdworkers, where multiple candidate documents are ordered according to their usefulness. The questions are open-ended and require deeper understanding rather than simple fact retrieval, making the ranking task more challenging.

Each dataset provides *human-labeled* judgments as a reliable reference. To complement these, we collect *LLM-generated* judgments following the judging principles outlined in the respective papers, ensuring consistency in evaluation criteria.

**LLM Selection.**  To obtain LLM-generated judgments, we employ a diverse set of both closed-source and open-source models across a wide range of sizes and model families. This diversity is essential to cover heterogeneous judgment patterns and to test detector generalization. Specifically, JD-Bench includes judgments from:

- **Closed-source models:**

Table 3: Overview of datasets included in JD-Bench.

| Dataset | HelpSteer2 | HelpSteer3 | NeurIPS | ANTIQUE |
|---|---|---|---|---|
| Application | Resp. Eval. | Resp. Eval. | Peer Review | Doc Ranking |
| Judgment Type | Pointwise | Pairwise | Pointwise | Listwise |
| Judgment Dims | Helpfulness, Correctness, Coherence, Complexity, Verbosity | Overall | Overall, Confidence, Soundness, Presentation, Contribution | Relevance |
| Rating Scale | 0–4 | –3–3 | 1–10 / 1–5 / 1–4 | 1–4 |
| #Train / #Test | 62,961 / 21,778 | 62,880 / 42,317 | 63,210 / 62,664 | 102,417 / 61,909 |

- – OpenAI series: GPT-4o, GPT-5-mini, o4-mini.
  - – Anthropics series: Claude-Haiku-3.5, Claude-Haiku-3, Claude-3.5-Sonnet.
  - – Google series: Gemini-2.0-Flash, Gemini-2.5-Flash, Gemini-1.5-Pro.
- **Open-source models:**
  - – LLaMA family: LLaMA-3.2-3B, LLaMA-3.1-8B, LLaMA-3.1-70B.
  - – Qwen family: Qwen-2.5-7B, Qwen-2.5-72B, RM-R1, RISE-Judge.
  - – Mistral family: Mistral-7B, Mistral-Small-24B.
  - – DeepSeek series: DeepSeek-V3, DeepSeek-R1.

This mixture of datasets and models results in a benchmark that is both large-scale and diverse: JD-Bench covers *multiple application scenarios*, *different judgment types* (score, pairwise, listwise), and *a wide spectrum of LLM families*, making it a comprehensive resource for advancing judgment detection research. Table 3 presents the statistics of JD-Bench.

**Prompt for JD-Bench Construction**

```
HelpSteer2 Prompt (Pointwise, 5-Dimension Scoring)

Given a prompt and a response, follow the rubric to make a
judgment.

## Rubric:
Judge the response on five aspects: helpfulness, correctness,
coherence, complexity, and verbosity.
Assign each aspect a scalar score in [0, 4].

## Prompt: [PROMPT]

## Response: [RESPONSE]

Please output a valid JSON object using the following
schema:     "Rationale": <explanation for the given scores>,
"Helpfulness": <0-4>,  "Correctness": <0-4>,   "Coherence":
<0-4>,  "Complexity": <0-4>,  "Verbosity": <0-4>

Formatted the abovementioned schema and produce the judgment JSON
now.
```

```
HelpSteer3 Prompt (Pairwise Comparison)

Given a prompt and two responses, follow the rubric to make a
comparative judgment.

## Rubric:  Compare Response 1 and Response 2 along five aspects:
helpfulness, correctness, coherence, complexity, and verbosity.
Assign a single comparative score in -3,-2,-1,0,1,2,3 using the
scale:   -3:  R1 much better than R2;   -2:  R1 better than R2;
```

```
-1: R1 slightly better than R2;    0: about the same;   1: R2
slightly better than R1;  2: R2 better than R1;   3: R2 much
better than R1.

## Prompt (conversation/context): [CONTEXT AS FLATTENED TEXT]

## Response 1: [RESPONSE_1]

## Response 2: [RESPONSE_2]

Please output a valid JSON object using the following schema:
"Rationale": <explanation for the comparative score>,  "Score":
<-3|-2|-1|0|1|2|3>

Formatted the abovementioned schema and produce the judgment JSON
now.
```

### NeurIPS Review Prompt (Structured JSON Review)

```
You are an AI researcher reviewing a paper submitted to a
prestigious AI conference.  Thoroughly evaluate the paper, adhering
to the provided guidelines, and return a detailed assessment in the
specified JSON format.

## Manuscript:  [MANUSCRIPT TEXT OR CONCATENATED CHUNKS]

## Reviewer Guidelines (dimensions to cover):
Summary:  Briefly summarize contributions (no critique here).
Strengths & Weaknesses across:  Originality, Quality, Clarity,
Significance.
Provide Questions for authors (useful for rebuttal).
Discuss Limitations and potential societal impact.
Flag Ethical Concerns if applicable (per conference policy).
Assign numerical ratings:  Soundness, Presentation, Contribution
(1-4 each).
Provide an Overall score (1-10) and Confidence (1-5).

## Output a valid JSON object with the following fields:
"Summary": <summary for the paper>,  "Questions": <questions
for the author>,  "Limitations": <limitations for the paper>,
"Soundness": <1-4>,  "Presentation": <1-4>,  "Contribution":
<1-4>,  "Overall": <1-10>,  "Confidence": <1-5>

Formatted the abovementioned schema and produce the review JSON
now.
```

### ANTIQUE Prompt (3-Way Relevance Ranking)

```
Given a prompt and three responses, follow the rubric to assess
relevance and rank the responses.

## Rubric (per-response relevance score in [1, 4]): 4:  Reasonable
and convincing; on par with or better than a likely correct
answer. 3:  Possibly an answer, but not sufficiently convincing;
a better-quality answer likely exists. 2:  Not an acceptable answer;
unreasonable or does not address the question, but still on-topic.
1:  Completely out of context or nonsensical.

## Prompt:  [QUERY]

## Response 1:  [RESPONSE_1]

## Response 2:  [RESPONSE_2]
```

```
## Response 3:  [RESPONSE_3]

Please output a valid JSON object using the following schema:
"Rationale":  <explanation for your judgment and ranking>,
"Response1 Score":  <1-4>,  "Response2 Score":  <1-4>,  "Response3
Score":  <1-4>,  "Ranking":  <list of indices indicating
best→worst, e.g., [0,1,2]>

Formatted the abovementioned schema and produce the judgment JSON
now.
```

## B.3  J-DETECTOR DETAILS

## B.4  LINGUISTIC FEATURES

We extract a comprehensive set of surface, lexical, syntactic, and discourse indicators from each candidate response using `spaCy`-based parsing pipelines.

- **Length & Structure:** `word_count`, `char_count`, `sentence_count`, `avg_sentence_length`, `list_count` (bullet or numbered lists), `paragraph_count`, `punctuation_count`, `reference_count` (e.g., URLs).

- **Lexical Diversity:** `unique_words`, `vocab_diversity` (unique/total word ratio), `average_word_length`, `noun_verb_ratio`, `adjective_ratio`, `adverb_ratio`, `pronoun_ratio`, `contraction_rate`.

- **Readability:** `coleman_liau` index.

- **Syntactic Complexity:** `syntax_tree_depth` (maximum dependency depth), `average_dependency_length`, `passive_voice_ratio` (fraction of sentences with `nsubjpass`/`csubjpass`), `subordinate_clause_rate` (rate of `mark` tokens).

- **Discourse/hedging:** `hedging_frequency` (occurrence of hedge words such as "may", "possibly"), `discourse_marker_rate` (connectives such as "however", "moreover").

These features are computed for each response independently. For pairwise or listwise datasets (e.g., HelpSteer3, ANTIQUE), we additionally compute *difference features* such as $r_1 - r_2$ on each scalar dimension when comparing two responses.

## B.5  LLM-ENHANCED FEATURES

Beyond surface-level indicators, we harness powerful large language models (e.g., Qwen3-8B) to derive task-aligned evaluation features. For each dataset, the model is prompted with the original instruction or query together with its candidate responses, and asked to generate structured JSON judgments that include detailed rationales and aspect-specific scores.

**Pointwise Setting (e.g., HelpSteer2).** Each response is scored independently along eight stylistic and content dimensions:

- Style, Format, Wording

- Helpfulness, Correctness, Coherence

- Complexity, Verbosity

The model outputs both a natural language rationale and numeric scores (0–4) per dimension plus an `overall_score`.

**Pairwise Setting (e.g., HelpSteer3).** Two responses are jointly compared under criteria such as *helpfulness*, *correctness*, *coherence*, *complexity*, and *verbosity*. The LLM produces a signed comparison score from $-3$ (Response 1 $\gg$ Response 2) to $+3$ (Response 2 $\gg$ Response 1) and a supporting rationale.

**Listwise Setting (e.g., ANTIQUE).**    Three responses are simultaneously ranked by relevance. The LLM assigns a 1–4 relevance score to each response and outputs an ordered ranking list $[0, 1, 2]$ to indicate relative quality.

**Long-form Paper Evaluation (e.g., NeurIPS Submissions).**    For full papers, we ask the model to return review-like signals: style, format, wording (0–4), rating (1–10), confidence (1–5), soundness/presentation/contribution (1–4 each), together with detailed reasoning.

Table 4: Example LLM-enhanced feature dimensions by dataset.

| Dataset Setting | LLM-Generated Feature Dimensions |
|---|---|
| HelpSteer2 (pointwise) | Style, Format, Wording, Helpfulness, Correctness, Coherence, Complexity, Verbosity, Overall |
| HelpSteer3 (pairwise) | Helpfulness, Correctness, Coherence, Complexity, Verbosity, Pairwise Score $(-3 - +3)$ |
| ANTIQUE (listwise) | Response relevance scores (1–4), Ranking order, Rationale |
| NeurIPS (pointwise) | Style, Format, Wording, Rating (1–10), Confidence (1–5), Soundness, Presentation, Contribution |

These LLM-enhanced features provide semantically rich, high-level signals that complement the surface-level linguistic statistics, enabling our detector to exploit both human-interpretable cues and task-specific, model-derived evaluations.

### B.6   SLM-BASED METHOD DETAILS

To benchmark the ability of small language models (SLMs) to discriminate between human and LLM-generated judgments, we adapt text classification pipelines with two input configurations: *judgment-only* (w/o candidates) and *judgment+candidate* (w/ candidates). Both settings train a binary classifier to predict whether a group of judgments originates from a human annotator (label 0) or an LLM (label 1). We employ `roberta-base` and `allenai/longformer-base-4096` as backbones, with max sequence lengths $512$ and $4096$, respectively.

- **Judgment-Only** Inspired by SLM-based text detection, this setting feeds only the *judgment artifacts* into the model. Each group is represented by a textualized summary of available signals, including:
    - *Numeric scores*: fields such as `rating`, `score`, `confidence`, `soundness`, `presentation`, `contribution`, etc.
    - *Pairwise comparisons*: keys such as `pairwise`, `pairs`, `comparisons`, or `prefs`.
    - *Ranking lists*: an explicit `ranking` field if available.
    - *Metadata*: optional question/prompt/task descriptions to provide minimal context.

  The resulting text is tokenized and directly used as the classifier input.

- **Judgment + Candidate** In this richer setting, we augment the above judgment text with the *candidate contents* being judged. Candidate responses are extracted from dataset fields such as:
    - `examples[*].docs` for passage-style corpora (e.g., ANTIQUE);
    - `examples[*].context` for conversational datasets (e.g., HelpSteer3), where only assistant turns are kept;
    - top-level `docs`, `candidates`, or `answers` if present.

  Since candidate texts can be long, we apply a *head+tail trimming* strategy per candidate to respect the model's maximum input length. Judgment tokens are prioritized to remain intact. The final input is a concatenation:

  $$\texttt{JudgmentText} \,\|\, \texttt{=== Candidates ===} \,\|\, \texttt{Candidate}_1 \,\|\, \ldots \,\|\, \texttt{Candidate}_n.$$

| Mode | Input Composition | Example Fields Used |
|---|---|---|
| w/o candidates | Judgments only | ratings, scores, pairwise, ranking, task |
| w/ candidates | Judgments + trimmed candidate texts | docs, context (assistant turns), answers |

Table 5: Two input modes for SLM-based judgment detection.

During training, both settings use the HuggingFace `Trainer` with standard hyperparameters (`AdamW`, learning rate $2 \times 10^{-5}$, batch size 8, weight decay 0.01). Labels are mapped to $\{0, 1\}$, with Human$\mapsto 0$ and LLM$\mapsto 1$. Evaluation reports accuracy, F1, and AUROC on held-out test splits.

### B.7 LLM-BASED METHOD DETAILS

### B.8 LLM-AS-A-JUDGE DETECTOR

Inspired by logits-based AI-generated text detection (Mitchell et al., 2023), we design a **single-pass detector** that treats an LLM as a surrogate judge. Given a group of judgments $G$, we build a compact textual payload including:

- **Judgment-only signals:** helpfulness, correctness, coherence, complexity, verbosity, ranking, and pairwise preferences.
- **Optional candidates:** trimmed prompt/response or passage text to provide weak context.

We prompt the detector LLM with an instruction template asking it to decide whether the judgments were written by a *Human* or by an *LLM*, based on style, consistency, and calibration artifacts:

```
{
  "Rationale": "<brief explanation>",
  "Prediction": "Human" | "LLM"
}
```

Two modes are supported:

- `judgment_only`: only judgment artifacts are provided.
- `enable_candidate`: judgment artifacts plus trimmed candidate texts.

This baseline does not use any explicit feature engineering but leverages the LLM's implicit ability to reason about stylistic and distributional cues.

### B.9 SAMPLE-LEVEL LLM-BASED ANALYSIS

We further design an **agentic feature mining** procedure to expose regularities in Human vs. LLM judgments at the *instance level*. Given a training set of groups, we:

1. Flatten them into a table of *prompt, response, label, scores*, and derived metrics such as length and average score.
2. Mine **Human–LLM pairs** using two strategies:
   - `scoring`: select $k$ pairs with the largest average-score gaps under the same prompt.
   - `pairwise`: sample $k$ random Human–LLM pairs.
3. Feed each pair to an LLM agent that proposes actions to maintain a **Feature Bank**:

```
Add:    {"name": "...", "description": "..."}
Delete: {"name": "..."}
Merge:  {"name": "...", "description": "...", "existing": "..."}
```

4. Typical mined features include:
   - Length or verbosity bias;
   - Overly smooth or formulaic score patterns;
   - Deterministic tone and calibration artifacts.

The resulting **Feature Bank** $\mathcal{F}_{\text{sample}}$ captures diagnostic cues distilled by the LLM itself and is later injected into the final detection prompt.

### B.10 DISTRIBUTION-LEVEL LLM-BASED ANALYSIS

Beyond individual samples, we analyze **dataset-wide statistics** to extract global signals of LLM-generated judgments:

1. Compute per-label histograms and descriptive statistics for all available judgment dimensions (e.g., helpfulness, correctness, coherence, complexity).

2. Analyze **correlations**:

   - Length–score Spearman correlations within Human/LLM groups;
   - Cross-dimension correlations (e.g., helpfulness vs. coherence).

3. Summarize these findings as structured text and feed them to an LLM to propose additional high-level features, such as:

   - Consistent score calibration (LLM often shows smaller variance);
   - Stronger length–score coupling in LLM judgments;
   - Reduced inter-dimension diversity compared to human raters.

The discovered global patterns augment the feature bank as $\mathcal{F}_{\text{dist}}$, complementing sample-level cues with distributional regularities.

### B.11  FINAL DETECTION

The final detector integrates:

- A **Feature Bank** $\mathcal{F} = \mathcal{F}_{\text{sample}} \cup \mathcal{F}_{\text{dist}}$;
- Group-level summaries (judgments + optional candidates).

An LLM receives this structured prompt and outputs the final label prediction:

$$\hat{y} = f_{\text{LLM}}(\text{summary}(G), \mathcal{F}),$$

where $f_{\text{LLM}}$ denotes the LLM-based reasoning process conditioned on both the mined features and the group payload.

Table 6: Comparison of the three LLM-based detection strategies.

| Method | Uses Candidates? | Feature Bank | Level of Analysis |
|---|---|---|---|
| LLM-as-a-Judge | Optional | None | Per-group |
| Sample-level | Optional | $\mathcal{F}_{\text{sample}}$ | Instance-level |
| Distribution-level | Optional | $\mathcal{F}_{\text{sample}} + \mathcal{F}_{\text{dist}}$ | Global + per-group |

In practice, the multilevel detector (sample + distribution) consistently improves accuracy by guiding the LLM with both fine-grained instance cues and global dataset regularities.

## C  THEORETICALLY ANALYSIS ON LLM-GENERATED JUDGMENT DETECTABILITY

We model the detectability of whether a group of judgments $G$ (scores, pairwise preferences, or listwise rankings) was produced by a human or an LLM. Let $m$ denote the group size, $d$ the number of attribute dimensions, and $S$ the effective rating scale cardinality:

$$S = \begin{cases} L, & \text{for } L\text{-level scoring;} \\ 2x + 1, & \text{for pairwise judgments with } x \in \mathbb{Z}_{\geq 1} \text{ superiority levels per side (including tie);} \\ k!, & \text{for a full ranking over } k \text{ candidates.} \end{cases}$$

The per-judgment information is $\log S$ nats.[2]

Let $P_H$ and $P_M$ be the conditional distributions over judgment outcomes induced by humans and LLMs, respectively. Denote $\Delta = \text{TV}(P_H, P_M)$ as their total variation distance.

---

[2]For listwise $k!$, Stirling's approximation gives $\log(k!) \approx k \log k - k$. For continuous pairwise margins, discretization into $B$ bins yields $S = B$.

**From sample complexity to group detectability.** With $n$ i.i.d. observations, the total variation between product distributions grows as

$$\text{TV}\big(P_H^{\otimes n}, P_M^{\otimes n}\big) = 1 - \exp\{-nI_c(P_H, P_M) + o(n)\},$$

where $I_c$ is the Chernoff information, scaling quadratically with $\Delta$. In our setting, the effective observation budget is

$$n_{\text{eff}} = m \cdot d \cdot \log S,$$

which accounts for group size, dimensionality, and rating resolution.

**Detectability index.** Thus, the detectability index becomes

$$\text{Det}(G) = 1 - \exp\{- \beta \, md \log S \, \Delta^2\},$$

where $\beta > 0$ is dataset- and model-dependent. The detectability increases monotonically with four factors: (i) rating scale $S$, (ii) attribute dimensions $d$, (iii) group size $m$, and (iv) distribution gap $\Delta$.

**Instantiation by type.** For $L$-level scores, use $S = L$. For pairwise preferences, use $S = L_{\text{pair}}$ (e.g., 7 for {-3,...,3}). For listwise ranking over $k$ items, use $S = k!$ (or $\log S \approx k \log k - k$). For mixed-type groups, sum $md \log S$ across instances.

