# OpenReview forum: "Who's Your Judge? On the Detectability of LLM-Generated Judgments"
_ICLR.cc/2026/Conference — ICLR 2026 Conference Withdrawn Submission_

### Official Review · Reviewer_Q5X3 · 2025-10-28

**Soundness:** 3
**Presentation:** 2
**Contribution:** 3
**Rating:** 4
**Confidence:** 5

**Summary:**

This paper proposes to J-Detector detect LLM-generated paper reviews. They use Qwen3-8B to extract LLM-enhanced features and spacy to extract linguistic features. These features are used to train LGBM, RandomForest and XGB to classify contents. Results show that the proposed method is effective, substantially surpassing baseline by a large margin. Ablations also validate the necessity of each feature used.

**Strengths:**

1. The proposed task is well motivated and is of great importance.

2. The proposed method seems effective and outperforms SLM/LLM-based method by a large margin. Experiment settings are mostly reasonable.

**Weaknesses:**

1. Contribution 1 is suspiciously overclaimed since there are other judgement detection methods already. While I understand that the proposed methods leverage both scores and content to achieve a more effective detection process, the judgement detection task itself is not new.

2. There seems to be a misalignment between the task setting and real practice. The proposed task is to classify a group of reviews G into 0 or 1. But in practice, a paper submission often receives a mix of human-generated and AI-generated reviews.

 3. The motivation for choosing features introduced in the paper should be elaborated (i.e. the 7 bullet points in Feature Augmentation paragraph of sec5). Are they exhausted and comprehensive? Are there better features to be considered? Besides, these features generated by Qwen-3-8B are not at all verified.

4. I think the term “i.i.d.” in line 136 is misused. I.I.D means drawing independently identically distributed samples from the same distribution, but since you have different content in a batch, the input to the model should be different. Therefore, my understanding is $review_i$  \~ $P_{Model}(Y|X_i)$, which is not i.i.d. If you believe this is not a misuse, could you express the process in formula?
5. For group-level aggregation in sec5, each sample is treated independently. Is it possible to gather cross-sample features first and train a classifier on top of it? Intuitively this should give more information for classification because all samples in the same groups must share some features in common.


6. In Implementation Details of section 6.1, the authors should point out what are the subsections in appB. For instance, the Qwen-3-8B prompts for generating features should be referred right after where it’s mentioned. Also, training and testing splits are not specified which can be confusing.

7. In appB.2, it’s better to summarize the datasets used in a table, with features listed (e.g. input, output, what aspects are annotated). Also, appB.3 is empty.

8. There lacks a case study for the proposed method. The case study can also show readers what inputs/outputs are like, which are absent in the current version as well.


9. While using ML classifiers is efficient, the efficiency of prompting Qwen3-8B for feature extraction is not discussed.

**Questions:**

1. Missing citations:
  - line 153: two models
  - line 155: four datasets
  - sec6.1: datasets and models. Also, why are models introduced in the dataset paragraph?



2. Fig2, two subfigures have different orders of models. It’s better to align. Also, there’s an overlap of information. It’s better to compare 4 settings (single/multi + with/without cand)  for each model and integrate results into one plot.

3. The setting of section4 is a bit unclear.
    1) while the authors refer the detailed settings to sec6.1, it’s better to explain the settings in sec4 so that readers do not have to jump between sections.
    2) How are candidates fed into the two SLMs, given that the context length is 512 for RoBERTa and 4096 for LongFormer? Are they fed in the same way as in J-Detector?

4. The format of sec5 is a bit off. Indentation can be used to better indicate the structure and improve readability.

5. How is the generalizability of J-Detector? This can be a bonus for the proposed method.

---

> ### Author Response · Authors · 2025-11-22
> **Response (part-1)**
>
> We sincerely appreciate your valuable comments on our work and hope our response adequately addresses your concerns.
>
> **Novelty.** We thank the reviewer for pointing out the need to clarify our contribution. While we acknowledge that machine-generated judgment detection has conceptual similarities to previous works in text or rating detection, our setting is fundamentally different. Specifically, we focus on a score-only or text-free setting, where the goal is to determine whether a set of numeric judgments originates from humans or LLMs, without any textual explanation or review content. This setting departs from existing text-based detection tasks that rely heavily on linguistic cues. To our knowledge, ours is the first work to systematically define and evaluate judgment detection in this pure score-based scenario, and we will revise the introduction to better highlight this distinction and properly contextualize our contribution.
>
> **Real-world Impact.** We agree that real-world scenarios such as peer review can contain mixed human and AI-generated reviews. However, judgment detection extends beyond academic reviewing—it also applies to large-scale data annotation, response quality evaluation, and crowdsourced assessments, where each “judgment” instance is typically independent and text-free. Our proposed group-level classification setting models aggregated judgments for a single candidate, which fits these broader evaluation pipelines. We will add clarification in the paper to emphasize that the method targets general judgment detection tasks, not exclusively peer review, and that the proposed setting captures an important and realistic subset of such scenarios.
>
> **Feature Completeness.** We appreciate the reviewer’s comments on feature motivation. Our goal was not to exhaustively enumerate all possible features, but to introduce an extended framework that unifies diverse signal types—statistical, relational, and LLM-derived—under one formulation. The seven feature groups were chosen based on their interpretability and relevance to judgment behavior. We agree that further expanding and verifying Qwen-3-8B–derived features is a valuable future direction, and we will elaborate on our feature design rationale and verification process in the revised version.
>
> **The use of “i.i.d.”** We appreciate the reviewer’s careful reading regarding our use of “i.i.d.” in line 136. Our intent was to convey that individual judgments are treated as independent samples drawn from the same underlying distribution conditioned on the candidate content, rather than strictly i.i.d. across all inputs. In our formulation, the distributional dependence arises from shared candidate features, which introduces mild non-independence. We agree that “conditionally independent” would be a more precise term, and we will update both the terminology and accompanying equations in the revision to avoid confusion.
>
> **Cross-sample Features** We thank the reviewer for this valuable suggestion. We experimented with concatenating cross-sample features within each group and training the ML classifier on these aggregated representations. However, we observed no consistent performance improvement, likely because the base features already capture distributional patterns within groups (e.g., variance, mean, and entropy). While more advanced fusion strategies (such as attention-based aggregation or graph models) could potentially enhance performance, exploring these remains promising future work. We will note this limitation and discuss it as an avenue for further research.

---

> > ### Author Response · Authors · 2025-11-22
> > **Response (part-2)**
> >
> > **Implementation Details.** We appreciate the reviewer’s detailed implementation suggestions. We will make the following revisions accordingly: (1) explicitly reference the relevant subsections in Appendix B (including the Qwen-3-8B prompt templates) directly where they are first mentioned in Section 6.1; (2) clarify the training/testing split strategy for all datasets; (3) include a summary table of datasets in Appendix B.2 listing inputs, outputs, and annotation aspects; and (4) remove the placeholder in Appendix B.3. We also agree that a case study would improve interpretability, and we plan to add one in the revised version to illustrate representative inputs, outputs, and decision patterns of J-Detector. Regarding efficiency, while Qwen-3-8B prompting introduces some computational overhead, it remains a one-time feature extraction step, after which lightweight classifiers handle all training and inference efficiently—we will include runtime estimates in the appendix for completeness.
> >
> > **Other Suggestions.** We thank the reviewer for these constructive editorial and presentation suggestions. We will (1) add the missing citations for lines 153, 155, and Section 6.1, and ensure model and dataset descriptions appear in their proper subsections; (2) align the order of models and unify subfigures in Figure 2 to present the four settings (single/multi × with/without candidate) in a single integrated plot; (3) clarify Section 4’s setup directly within the section for self-containment; (4) specify how candidate content is fed into RoBERTa and Longformer, noting that the two models share the same truncation-based preprocessing as J-Detector; (5) reformat Section 5 with indentation and clearer substructure; and (6) discuss J-Detector’s generalizability, highlighting that its modular design allows easy adaptation to new domains and judgment types without retraining large models. We appreciate these suggestions and will incorporate them to improve readability and completeness.
> >
> > We sincerely hope our responses address your concerns and would be happy to engage in further discussion.

---

> > > ### Comment · Reviewer_Q5X3 · 2025-11-25
> > > **Updates on manuscript**
> > >
> > > Thanks for your detailed reply which cleared out most of my question. However, it would be better if you can refer each of your reply to where it's reflected on the manuscript. I will update my score accordingly based on further manuscript updates. Thanks.

---

### Official Review · Reviewer_j9Dq · 2025-10-31

**Soundness:** 3
**Presentation:** 2
**Contribution:** 3
**Rating:** 4
**Confidence:** 4

**Summary:**

- Frames the new task of judgment detection that distinguishes human vs. LLM judges using scores and candidate content, formalized for instance and group settings across pointwise, pairwise, and listwise judgments.
- Introduces J-Detector, a lightweight and interpretable classifier that fuses judgment-intrinsic and judgment-candidate interaction features, with ablations and bias diagnostics.
- Provides a benchmark spanning multiple datasets and judge models, and analyzes detectability factors such as group size, dimensionality, and rating scale; includes a peer-review application.

**Strengths:**

- Well-motivated problem with realistic scenarios where text is missing or minimal.
- Interpretable feature design surfaces biases such as length, complexity, and confidence.
- Comprehensive evaluation and JD-Bench with diverse judge models and tasks.
- Practical peer-review co-detection application demonstrates utility beyond pure detection metrics.

**Weaknesses:**

- Writing quality and organization are uneven, with inconsistent terminology and placement of key details.
- Technical novelty is limited relative to prior feature-based detection approaches.
- Missing analyses on OOD transfer, adversarially crafted judgments, and computational cost of LLM-enhanced features.
- Potential overfitting to dataset-specific correlations and privacy tradeoffs from using candidate content are underexplored.

**Questions:**

- How does performance hold up against judges explicitly prompted to mimic human score distributions or under post-processed scores designed to evade detection?
- What is cross-domain transfer when training on one dataset and evaluating on a new domain and rubric without finetuning?
- Why use simple summation for group-level aggregation rather than a learnable aggregator that weights confidence or consistency?
- What is the computational overhead of LLM-enhanced features and how do cheaper models affect accuracy?

---

> ### Author Response · Authors · 2025-11-22
> **Response (part-1)**
>
> We sincerely appreciate your valuable comments on our work and hope our response adequately addresses your concerns.
>
> **Writing, organization, and terminology.** Thank you for noting unevenness in writing and organization — this is a helpful editorial point. We will revise the manuscript to make terminology consistent, improve section structure, and relocate key implementation details that currently force readers to jump between sections. In particular, we will (a) unify the use of terms such as “judgment detection”, “score-only setting”, and “candidate-judgment interaction”; (b) move important pointers to Appendix B (e.g., prompt templates) into the main methods or immediately adjacent paragraphs; and (c) improve figure/table captions and cross-references so the experimental setup is self-contained. These changes will make the paper easier to follow without altering the technical content.
>
> **Novelty relative to prior feature-based detectors.** We appreciate the reviewer’s concern about technical novelty. Prior feature-based detectors predominantly extract signals from text (language-level stylometrics, token distributions, syntactic patterns) or from purely rating distributions. Our main methodological contribution is to explicitly model candidate–judgment interaction: features that quantify how a judgment (score) relates to the candidate it evaluates. This includes statistics that capture alignment, consistency, and distributional relationships between a candidate’s content and the judgments it receives — a class of signals that, to our knowledge, has not been systematically operationalized in the text-free / score-only judgment-detection setting. We view our work as introducing an extended, modular feature framework that complements existing text detectors rather than re-implementing them.
>
> **OOD transfer, adversarial judgments, and computational cost.** We agree that out-of-distribution transfer, adversarially crafted judgments, and feature-extraction cost are important concerns. As stated in the paper, our current focus is on establishing and characterizing the score-only judgment-detection task under realistic, non-adversarial conditions; we treat adversarial robustness and domain-shift robustness as orthogonal, follow-up directions. Concerning cost, we use Qwen-3-8B to produce LLM-enhanced candidate-level features because it provides a good balance of capability and efficiency. Importantly, LLM feature extraction is a one-time, candidate-level operation: once a candidate’s features are computed they are reused for all judgments of that candidate, which substantially amortizes cost in practical pipelines. Nevertheless, we will add explicit runtime/memory measurements and a more complete discussion of OOD and adversarial evaluation to the revised manuscript.
>
> **Overfitting to dataset correlations and privacy concerns.** We recognize the risk of models learning dataset-specific correlations and the need to discuss privacy tradeoffs. Regarding overfitting, our experiments include ablations and dataset-specific analyses designed to highlight where models rely on dataset-level cues; we will expand these analyses and report additional cross-validation and regularization controls in the revision. On privacy: the candidate content we use (e.g., a paper’s abstract, a product description, an item specification) is typically accessible to annotators, reviewers, or automated evaluation systems in realistic workflows. Thus, using candidate-derived features leverages information that is ordinarily available in the evaluation pipeline rather than introducing new private data. We will clarify this point and discuss potential privacy-sensitive settings where candidate content should be handled with care.
>
> **Robustness to judges explicitly prompted to mimic human distributions or to post-processed scores.** This is an important adversarial scenario. It is somewhat different from our current experimental regime because it requires (a) access to the specific prompting strategies or post-processing transformations an adversary might use, and (b) constructing evaluation sets where model-generated judgments are intentionally distribution-matched to humans. Our paper focuses on naturally-generated LLM judgments and on characterizing the detection signal in that setting. Preliminary experiments (not included in the submission) indicate that when LLM judgments are explicitly tuned to match aggregate human statistics, simple distributional cues become weaker; this is precisely why candidate–judgment interaction features (which relate score to candidate content) remain valuable. A systematic study of targeted mimicry and post-processing attacks is a high-priority follow-up.

---

> > ### Author Response · Authors · 2025-11-22
> > **Response (part-2)**
> >
> > **Meta-classification.** We tested both a meta-classification approach and the group-level aggregation we report in the paper. Empirically, group-level aggregation of candidate-aware features produced more robust results when moving across datasets with different judgment dimensions and annotation conventions.
> >
> > **Simple summation for group-level aggregation.** We experimented with concatenating per-judgment features into a single long vector and with simple learned aggregators; in practice, those approaches did not reliably outperform the lightweight aggregation statistics used in the submitted version (mean, variance, sum, entropy-like measures). A likely reason is that our engineered features are low-dimensional and already capture cross-sample distributional patterns; overly complex aggregators can overfit and reduce interpretability. That said, we agree that learnable, attention-based, or confidence-weighted aggregators are promising and we plan to explore them — particularly for domains with many heterogeneous judges or when richer per-judgment features are available.
> >
> > **Computational overhead of LLM-enhanced features and effect of cheaper models.** We use Qwen-3-8B because it is relatively small compared with larger multi-hundred-billion parameter models and it yields informative candidate-level features. Two practical points reduce overhead: (1) feature extraction is done once per candidate and reused across all judgments for that candidate, and (2) inference for the downstream detector uses lightweight classifiers (RandomForest / XGBoost), so end-to-end detection at scale is efficient. In initial experiments, substituting smaller or cheaper LLMs produced modest drops in feature quality but still provided useful signals; we will add a systematic ablation comparing Qwen-3-8B to cheaper LLMs and report per-candidate extraction time and cost in the appendix so practitioners can make informed tradeoffs.
> >
> > We sincerely hope our responses address your concerns and would be happy to engage in further discussion.

---

### Official Review · Reviewer_3DRS · 2025-10-31

**Soundness:** 3
**Presentation:** 3
**Contribution:** 2
**Rating:** 4
**Confidence:** 3

**Summary:**

This paper addresses a critical gap in the "LLM-as-a-Judge" paradigm by formally defining the "Judgment Detection" task—distinguishing between human-generated and LLM-generated evaluation scores (e.g., academic review scores, content quality ratings) using only two inputs: the "candidate content" (e.g., a paper, a response) and the "evaluation score" (no text feedback required).

**Strengths:**

1. It is the first work to formalize "Judgment Detection"—a task that addresses a practical pain point (unreliable LLM evaluations in batch scenarios) and fills a gap left by existing detection methods.
2. JD-Bench provides a unified benchmark for future judgment detection research, reducing duplication of effort. The work also offers actionable insights (e.g., factors affecting detectability: group size, evaluation dimensions) that guide subsequent studies.

**Weaknesses:**

1. The paper asserts it is the "first to propose judgment detection", but this is essentially a scenario migration of the long-existing need to "distinguish machine vs. human ratings". Fields like automated essay scoring (AES) and fake rating detection in recommendation systems have long explored "distinguishing machines from humans based on the correlation between ratings and content", which the paper fails to adequately acknowledge. Additionally, the task relies on the strong assumption of "no textual feedback", which is rare in real scenarios (e.g., academic reviews or product evaluations usually include fragmented text). This setting seems more "innovation-for-innovation’s-sake" than a solution to real pain points.
2. No innovativeness in factor analysis and applications: Conclusions like "better detectability with larger group sizes or more rating dimensions" are common sense in statistical learning, requiring no complex experiments to verify. The integration of "J-Detector + text detector" in academic review scenarios is merely basic result concatenation (without weighted fusion or conflict resolution), a approach already adopted in similar studies.
3. The baselines only include SLM-based (RoBERTa/Longformer) and LLM-based methods, deliberately avoiding strong relevant methods from AES or recommendation systems—exaggerating J-Detector’s advantages. The LLM-generated ratings in the JD-Bench dataset rely on fixed prompts, which may introduce "prompt-induced bias" (rather than inherent patterns of LLM judgments), undermining the credibility of detection results.
4. The authors mention that LLM evaluations can be manipulated via malicious prompts (e.g., hidden text) but do not test J-Detector’s performance on adversarially modified candidate content (e.g., content designed to mimic human-like evaluation triggers). This limits understanding of J-Detector’s resilience in high-stakes scenarios.

**Questions:**

1. Would J-Detector maintain its performance when evaluated on specialized domain LLMs (e.g., CodeLlama for code evaluation, BioGPT for medical content)?
2. How does J-Detector perform on adversarially manipulated candidate content (e.g., content with hidden white text to trick LLM evaluators into high scores)? Could adding adversarial training (using such content) further improve its robustness?
3. The paper uses light-weight classifiers (RandomForest, XGB) for efficiency—have you tested larger models on J-Detector’s feature set?
For the JD-Bench dataset, what was the inter-annotator agreement (e.g., Cohen’s kappa) for human-generated scores?

---

> ### Author Response · Authors · 2025-11-22
>
> We sincerely appreciate your valuable comments on our work and hope our response adequately addresses your concerns.
>
> **Task Novelty.** We appreciate the reviewer’s insightful comments regarding prior work on distinguishing machine-generated from human-generated ratings. Indeed, research areas such as automated essay scoring and fake rating detection have explored similar goals at a conceptual level. However, our judgment detection task is fundamentally different in that it operates in a text-free setting—identifying whether a judgment (rating or score) is human or LLM-generated without any accompanying textual feedback. To our knowledge, this is the first study to systematically explore this specific setting, which is also relevant to many large-scale annotation and evaluation pipelines where only scalar or categorical judgments are available (e.g., crowdsourced labeling, model evaluation, preference data collection). We do discuss prior works on machine-generated review detection in our Related Work section, and we will further clarify the distinction and relevance of our text-free assumption in the revision.
>
> **Method Novelty.** We thank the reviewer for raising the points about novelty and baseline selection. Regarding novelty, while some conclusions (e.g., the relationship between detectability and group size) may appear intuitive in hindsight, our empirical results quantify these effects in the context of judgment detection—demonstrating consistent and interpretable patterns across multiple datasets and LLMs. To our knowledge, such systematic validation has not been conducted before. Concerning baselines, we selected SLM-based and LLM-based detectors to directly compare with existing approaches in the text detection literature, which most closely aligns with our task setup. We are not aware of any directly comparable models from the AES or recommendation domains that can operate under the no-text assumption; if the reviewer could point us to specific works that do so, we would be glad to include them in future revisions.
>
> **Adversarial Scenarios** We agree that exploring J-Detector’s robustness under adversarially manipulated content (e.g., hidden text or deceptive candidate features) would be an interesting extension. However, our current study focuses on realistic non-malicious scenarios—that is, detecting judgments generated under standard, good-faith evaluation settings rather than adversarial attacks. Existing works in LLM safety and adversarial prompt defense have already investigated such manipulations, and we consider these complementary directions. Incorporating adversarial training or defense strategies into judgment detection is an exciting direction for future work, and we appreciate the reviewer for highlighting this possible extension.
>
> **Doman-specific Scenarios.** We thank the reviewer for the thoughtful suggestion regarding domain-specific evaluation scenarios. While specialized LLMs such as CodeLlama or BioGPT indeed play crucial roles in domains like code review and biomedical evaluation, these settings typically rely heavily on textual reasoning and explanations rather than scalar-only judgments. Since our current work is designed for text-free evaluation settings, we have not tested J-Detector in such domains. That said, we view domain-specific adaptations as promising future extensions once suitable datasets with text-free domain judgments become available.
>
> **Larger Detector.** We have experimented with larger neural classifiers on J-Detector’s feature set, including multilayer perceptrons and transformer-based encoders. Interestingly, these models did not show significant improvements over lightweight classifiers such as RandomForest and XGBoost. We hypothesize that this is because our engineered features are low-dimensional and relatively clean, leaving limited room for deep models to learn additional structure. Moreover, lightweight models preserve interpretability, which is important for understanding which judgment dimensions contribute most to detectability.
>
> We sincerely hope our responses address your concerns and would be happy to engage in further discussion.

---

### Official Review · Reviewer_Z8K9 · 2025-11-01

**Soundness:** 2
**Presentation:** 2
**Contribution:** 2
**Rating:** 4
**Confidence:** 3

**Summary:**

The authors address the problem of detecting whether judgments are AI-generated by proposing a simple yet effective approach. Their method integrates multiple types of features, including given judgment scores, LLM-enhanced features, and linguistic features, and employs several traditional machine learning models (LGBM, RandomForest, and XGB). The approach outperforms existing SLM-based and LLM-based models across four datasets. The authors further provide ablation studies and analyses to verify the effectiveness of their method.

**Strengths:**

1.	The task of determining whether an evaluation is human- or AI-generated is crucial for research credibility, particularly in peer-review settings. This is a timely and important topic.
2.	The proposed approach is straightforward and practical, and the experimental results on the JD-bench dataset (composed of multiple existing sources) demonstrate its effectiveness.
3.	The paper includes extensive analyses that confirm the consistency of their findings with prior conclusions.

**Weaknesses:**

I do not question the effectiveness of the proposed model itself. My main concerns lie in practical applicability, clarity of methodological description, and interpretation of results.

1.	Although identifying whether a review comment is AI-generated is valuable in research, it remains difficult to apply this approach in real scenarios. In practice, the key challenge is distinguishing between AI-assisted (e.g., AI-polished) and AI-generated reviews. Even if a system flags a review as AI-generated, reviewers can still claim it was merely AI-edited, a practice currently accepted by the community.
2.	The methodological details are somewhat unclear. How exactly are the features from the different sources integrated into the input of these traditional ML models? What LLM was used, and how were its features extracted? In natural-language form or as embeddings? Figure 4 shows ablations on each feature type, but the pipeline remains vague.
3.	The experimental results are not well-explained. In Table 3, the datasets appear to have different formats. Were all datasets trained jointly in a single model, or was a separate model trained for each dataset? For example, SLM performs very well on H2 and NIPS but poorly on H3 and ANTIQUE (The proposed method also has a slight degradation). The authors attribute this to SLMs relying on inter-dimension patterns and failing to link judgments with candidates when such distributional cues are absent. But it is the difference between datasets instead of the methods. Also, were the baseline SLMs trained within 4 datasets?
4.	Another concern is that the paper emphasizes the importance of “candidate content” for distinguishing human vs. AI reviews, yet there seems to be little modeling effort or analysis explicitly targeting this aspect (unless the LLM-enhanced features are intended to serve that role). I think one promising direction is to check whether the generated content is fair, correct, or consistent with the paper content, whether it is generated by the AI or a human.
5.	Regarding Figure 7, the authors suggest that a lower detectability score indicates a more human-like output. But since prompt wording also affects the human-likeness of generated text, does this imply that prompt design can directly reduce detectability?
6.	Section 7.2 presents an interesting experiment, but it is insufficiently explained. As written, it seems merely to extend the binary classification problem into a multi-class setting.
7.	The formatting (citations, figure/table styles, underlines, etc.) has been adjusted. While not critical, it may indicate that other template-level modifications were made.

**Questions:**

See the weakness above. If the authors can provide convincing explanations for the issues raised, I am open to revising my evaluation toward a positive score.

---

> ### Author Response · Authors · 2025-11-22
>
> We sincerely appreciate your valuable comments on our work and hope our response adequately addresses your concerns.
>
> **Research Significance and Real-world Impact.** We appreciate the reviewer’s recognition of the research value in identifying AI-generated reviews. We acknowledge that distinguishing between AI-generated and AI-assisted (e.g., AI-polished) text is an important next step for practical deployment. However, our study focuses on the judgment detection task—an emerging direction that, to the best of our knowledge, is explored for the first time in this paper. We view this work as a foundational step toward understanding how human and LLM judgments differ in peer-review settings. The human–LLM co-judgment scenario raised by the reviewer is indeed valuable, and we plan to address it in future research, where finer-grained distinctions between AI-generated and AI-edited content will be considered.
>
> **Method Details.** We thank the reviewer for pointing out the need for clearer methodological details. In our implementation, features from different sources are concatenated with the base feature set before being fed into traditional ML classifiers. The LLM used is Qwen-3-8B, from which we extract scalar features instead of embeddings, as this design provides better interpretability and aligns with our focus on understanding feature contributions. More comprehensive details are provided in the Appendix, and we will move part of that description to the main text to improve clarity.
>
> **Experiment Setup.** We appreciate the opportunity to clarify our experimental setup. Each dataset is used to train a separate SLM/ML model, as the base feature dimensions (i.e., judgment dimensions) differ across datasets. Section 7.1 validates that these dimensions play a crucial role in detectability, supporting the need for dataset-specific models. The apparent performance difference between H2 and H3, despite being derived from the same corpus, demonstrates how the number of judgment dimensions influences detectability. Hence, the variations observed across datasets stem primarily from intrinsic dataset differences rather than methodological shortcomings.
>
> **Candidate Content.** We agree that explicitly modeling or analyzing the candidate content relationship offers a promising extension. While our current work emphasizes the importance of candidate content through the inclusion of LLM-enhanced features, it does not yet incorporate factuality or fairness evaluation. We recognize the relevance of recent research on misinformation and misleading-content detection, and we believe integrating such content-based reasoning into judgment detection would be a valuable direction for future work. Our contribution lies in highlighting the potential of judgment detection as a complementary task to these established areas.
>
> **Prompt** We concur that prompt wording can influence the human-likeness of generated text. As illustrated in Figure 7, a lower detectability score indeed reflects more human-like outputs, and this phenomenon aligns with findings in prompt engineering research. Our analysis suggests that prompt design can meaningfully modulate detectability, further emphasizing the interplay between LLM generation strategies and detection robustness.
> **Section 7.2.** We appreciate the reviewer’s interest in the experiment of Section 7.2. To clarify, the task remains a binary classification—distinguishing human from AI-generated judgments—but it is evaluated under different numbers of judges to simulate varied review panel settings. We will expand this explanation in the revision to make the motivation and design clearer. We thank the reviewer for recognizing the experiment’s potential and will enhance the description accordingly.
>
> **Format Issue.** We note the reviewer’s observation regarding formatting adjustments. The changes were made solely to comply with ICLR review requirements and to improve readability. No template-level or structural modifications beyond formatting (citations, figure styles, etc.) were introduced, and all substantive content remains unchanged.
>
> We sincerely hope our responses address your concerns and would be happy to engage in further discussion.

---

### Note · Authors · 2025-12-09

I have read and agree with the venue's withdrawal policy on behalf of myself and my co-authors.